# SFCLTA: Spectral Fusion Contrastive Learning with Topology-Adaptive Graph Augmentation

Zhuo Xu [1]  Lu Bai [1]  Jincheng Li [1]  Lixin Cui [2]  Ming Li [3]  Hangyuan Du [4]  Yue Wang [2]

## Abstract

Graph Neural Networks (GNNs) have achieved remarkable success in graph analysis due to the Message-Passing (MP) mechanism, yet they struggle with heterophilic graphs where connected nodes often have distinct labels or dissimilar attributes. Graph Contrastive Learning (GCL) serves as a promising approach to extract the information beyond neighboring nodes, effectively mitigating the limitations of the MP mechanism in handling heterophilic graphs. Nevertheless, GCL faces two critical challenges when applied to heterophilic graphs, i.e., the potential distribution shift from data augmentation and the loss of robustness caused by high-frequency signals. To address these problems, we propose a novel model, namely the Spectral Fusion Contrastive Learning with Topology-Adaptive Graph Augmentation (SFCLTA) for unsupervised graph representation learning. Our method dynamically adjusts graph structures via a heterophily-aware augmentation strategy, and constrains high-frequency distortions by spectral regularization. We utilize the confidence-weighted fusion to enhance the robustness. Additionally, we introduce a feature reconstruction task as a prerequisite to explicitly mitigate feature-level distribution shifts. Experiments on multiple real-world datasets demonstrate that the proposed SFCLTA consistently outperforms baseline models in multiple tasks. Our code is released on https://github.com/JonathanGXu/SFCLTA.

[1]School of Artificial Intelligence, Beijing Normal University, Beijing, China. [2]School of Information, Central University of Finance and Economics, Beijing, China. [3]Zhejiang Key Laboratory of Intelligent Education Technology and Application, Zhejiang Normal University, Jinhua, China. [4]School of Computer and Information Technology, Shanxi University, Taiyuan, China. Correspondence to: Lixin Cui <cuilixin@cufe.edu.cn>.

*Proceedings of the $43^{rd}$ International Conference on Machine Learning*, Seoul, South Korea. PMLR 306, 2026. Copyright 2026 by the author(s).

## 1. Introduction

### 1.1. Background

GNNs associated with the MP mechanism (Luo et al., 2024) are powerful tools for analyzing graph data in many real-world applications. The success of the MP mechanism is commonly attributed to the homophily principle (McPherson et al., 2001), and this principle implies that individuals with similar attributes tend to connect with each other when aggregating the information from neighboring nodes. Under this assumption, the MP mechanism encourages the similarity among intra-class features and enhances the distinctiveness of inter-class node representations. Consequently, GNNs often outperform traditional neural networks on node-level tasks (Kipf & Welling, 2016a; Bai et al., 2022; Zheng et al., 2022; Bai et al., 2023; Hua et al., 2024).

However, the homophily assumption often breaks down in real-world graphs. This is because multiple graphs including technological graphs and biological graphs are inherently heterophilic in topological structure (Newman, 2003). Heterophily indicates that connected nodes frequently belong to different classes or have dissimilar features (Lozares et al., 2014). Since heterophilic edges are the bridge between nodes with different classes, the aggregated node representations become indistinguishable due to the MP mechanism (Luan et al., 2022). As a result, GNNs associated with the MP mechanism tend to suffer significant performance degradation on node-level tasks on heterophilic graphs.

To tackle the issue of performance degradation in GNNs when processing heterophilic graphs, GCL has emerged as a mainstream approach. As a self-supervised method, GCL learns representations that capture features beyond the local aggregation typically performed by GNNs (Ju et al., 2024). Generally, a typical GCL follows two key steps including **augmentation** and **encoding**, i.e., GCL generates augmented graphs according to the input graph, and learns representations by contrasting the augmented graph views encoded with a GNN-based encoder. Specifically, the augmentation mechanism allows GCL to capture the information for distinguishing nodes when coping with heterophilic graphs. In particular, the richness of graph information has led several GCL methods (Ren et al., 2023;

Li et al., 2024a; Shou et al., 2025) to adopt graph signal decoupling approaches. In these methods, low-pass filters (LPFs) extract the low-frequency components of graph signals, representing global views, while high-pass filters (HPFs) capture the high-frequency components, representing local views (Sandryhaila & Moura, 2014). The two components are then encoded separately, and contrastive losses are computed accordingly.

Researchers further leverage graph signals for better learning graph representations with multi-frequency signals within the GCL framework. As shown in the previous studies on spectral graph theory, GNNs are usually considered as LPFs to capture graph low-frequency signals (Nt & Maehara, 2019), which in turn helps to emphasize the global features. Also, the high-frequency signals are critical in recognizing the graph heterophily patterns (Zhu et al., 2020). Motivated by this observation, numerous methods leverage the multi-frequency components to improve representation learning. But there are several notable challenges in the existing studies. We summarize these challenges as follows.

**(1) Potential Distribution Shift from Data Augmentation.** In the contrastive learning framework, data augmentation explicitly changes the original graph topology to generate a diverse view. However, such topological changes without constraints may disturb the high-frequency signals, which are more sensitive to structural alterations than the low-frequency ones (Rossi et al., 2024). The traditional preprocessing graph augmentation methods or graph structure learning methods cannot perceive the impact of topology changes on high-frequency signals, resulting in a large ***distribution shift*** during the training process. As a result, the high-frequency signals degenerate into noise, leading to semantic collapse and prediction bias on downstream tasks such as node classification (Luan et al., 2023). Although existing studies (Zhu et al., 2021b; Zhang et al., 2021) propose adaptive augmentation strategies, they mainly optimize the semantic augmentation and lack consideration for the spectral properties of graph topology. These methods cannot ensure that the augmented graph maintains consistency with the original graph in the spectral domain, especially for the high-frequency signals. The distribution shift is not fundamentally controlled.

**(2) The Loss of Robustness Caused by High-Frequency Signals.** The HPF utilized in extracting the high-frequency signals suffers from ***weak robustness*** (Chang et al., 2021). In GCL, the HPF is vulnerable to the noise and perturbations from the augmentation (Dabush & Routtenberg, 2024). These issues both limit the effectiveness of the current multi-frequency contrastive learning on heterophilic graphs. Existing GCL methods that utilize multi-frequency signals (Chen et al., 2024; Yang & Mirzasoleiman, 2024) recognize the importance of HPFs, but they directly utilize the features

extracted by HPFs for contrastive learning. They lack a mechanism to distinguish between the high-frequency signals containing reliable discriminative information and the noise introduced by the augmentation process.

The details and related proofs of these challenges are introduced in Appendixes A.2 and A.3.

### 1.2. Contributions

To address these problems, we propose a novel **Spectral Fusion Contrastive Learning with Topology-Adaptive Graph Augmentation (SFCLTA)** for unsupervised graph representation learning. SFCLTA comprises two primary components, i.e., **Topology-Adaptive Graph Augmentation (TAGA)** and **Spectral Fusion Contrastive Learning (SFCL)**. Both of them aim to protect and robustly utilize high-frequency signals in GCL for heterophilic graphs. TAGA controls the augmentation process at the graph structure level. SFCL processes a full spectrum of signals, encompassing both low and high frequencies, for representation learning. Unlike GCL methods relying on multi-frequency signals, spectral fusion methods, which are widely adopted in hyperspectral image classification, not only extract multiple signals, but also integrate these signals to generate fusion representations (Wan et al., 2025). Spectral fusion methods capturing both global and local features effectively mitigate dependence on a single signal and reduce sensitivity to noise (Sun et al., 2022; Wang et al., 2023). We transfer the spectral fusion to GCL in SFCL, which extracts and fuses low- and high-frequency signals from the graph. Furthermore, we discuss the spectral fusion in Sections 3.5 and 4.2. We list our contributions as follows.

**Firstly**, in the TAGA, we adopt the strategy that dynamically modifies the graph structure according to node features and global heterophily, and introduce a spectral regularization as part of the loss function. The TAGA approach allows the generation of the augmented graphs with controllable changes, reducing destructive interference with the high-frequency signals. Meanwhile, to improve the loss function, we add a spectral regularization term that explicitly constrains the distribution shift caused by the graph augmentation. Compared to the semantic augmentation methods, the generated augmented views are more spectrally stable. Also, we explain that the proposed method can bound the KL-divergence between the distributions of high-frequency signals before and after augmentation within an upper limit (cf., Theorem 1 in Section 3.4).

**Secondly**, the SFCL paradigm integrates multi-frequency signals. Different from traditional contrastive learning based on the low-frequency signals (Bo et al., 2021), we argue that both low-frequency (global similarity) and high-frequency (local dissimilarity) features are essential for effective learning on the heterophilic graphs. To this end, we extract multi-

frequency signals from both the original and the augmented graphs and then adopt the improved spectral fusion method. To reduce the sensitivity of the HPF to noise, we develop a confidence-weighted fusion mechanism that adaptively integrates high-frequency features in the multi-frequency fusion process. With the weighted contribution of each high-frequency signal, this mechanism reduces the impact of noisy and misaligned features. As shown in Theorem 2, we further point out that this mechanism stabilizes the training process of contrastive learning by suppressing the gradient contribution of noisy nodes in theory. Moreover, we introduce a feature reconstruction pretext task that facilitates feature alignment in the semantic space. This pretext task further stabilizes contrastive training and especially avoids the distribution shift of features.

**Finally**, we evaluate the performance of our proposed model on a variety of homophilic and heterophilic graph datasets in experiments, demonstrating the effectiveness of the proposed SFCLTA. On node-level and graph-level tasks, our proposed model exhibits excellent performance. To further explore the proposed SFCLTA, we also set ablation experiments and sensitivity analysis.

Next, we first introduce related works in Section 2. Then, we describe the proposed model in Section 3. Experiments and related analyses are presented in Section 4. Finally, we summarize our work in Section 5.

## 2. Related Works

### 2.1. Graph Neural Networks

GNN is a popular neural network, which has been widely applied in many scenarios (Bai et al., 2022; Wu et al., 2022; Jin et al., 2023; Gao et al., 2023). In analyzing graph data, GNNs play an important role due to the message passing mechanism. In detail, the message passing process could be divided into 3 steps: aggregating, combining and readout.

For example, Graph Convolutional Network (GCN) (Kipf & Welling, 2016a), a typical kind of GNNs, extracts discriminative features for the graph classification task. Given an input graph $G = (X, A)$ with the feature matrix $X$ and the adjacent matrix $A$, the embedding matrix $Z$ is computed as

$$Z = \text{GCN}(X, A) = \text{ReLU}(\tilde{D}^{-\frac{1}{2}} \tilde{A} \tilde{D}^{-\frac{1}{2}} XW), \quad (1)$$

where $W$ is a trainable matrix, $\tilde{A} = A + I$ is the adjacent matrix with the self loop, $\tilde{D} = \sum_j \tilde{A}_{ij}$ is the corresponding degree matrix.

With the comprehension of the graph data and the GNN methods, the relationship between graph structure information and node features became the focus in recent work (Luan et al., 2023; Rossi et al., 2024; Khemani et al., 2024). The homophily principle, i.e., the nodes with similar attributes tend to be more connected, is commonly thought of as the main reason for the effectiveness of GNNs (Luan et al., 2024). However, in many cases (Luan et al., 2023), the GNNs with the homophily principle do not have an excellent performance. The heterophily (i.e., low homophily) is considered a necessary factor in the design of models (Yang & Mirzasoleiman, 2024).

**Graph Signal Processing.** Graph signal processing derives from the need to apply the signal processing techniques to non-Euclidean data. Early developments are influenced by spectral and algebraic graph theory. Based on the classical Fourier analysis, graph signals could be transferred into linear combinations of a basis of graph signals, where the basis vectors are ordered according to the frequency (Leus et al., 2023). The operator that captures the frequency with respect to the undirected graph is the normalized Laplacian matrix $L = I - D^{-\frac{1}{2}} A D^{-\frac{1}{2}}$. Since $L$ is a real symmetric matrix, the Laplacian could be represented as $L = U\Lambda U^\top$, where $\Lambda = \text{diag}(\lambda_0, \cdots, \lambda_{n-1})$ represent ascending order eigenvalues $\{\lambda_i\}_{i=0}^{n-1}$, and $U = [u_0, \cdots, u_{n-1}]$ are the corresponding eigenvectors. The graph signal filter is based on the graph Fourier transform. From the graph filter view, the GNN with Equation 1 is a typical LPF (Li et al., 2021).

### 2.2. Graph Self-supervised Learning Methods

The self-supervised learning (SSL) for the graph data has been explored, and various methods (Veličković et al., 2019; Hou et al., 2022; 2023) based on SSL have been proposed and applied (Liu et al., 2022). We introduce the two main frameworks of the graph SSL in the following contexts.

**Graph Auto-Encoders (GAEs).** Similar to the Auto-Encoder framework (Vaswani et al., 2017), the GAEs usually combine with the encoder and the decoder. Given an input graph $G = (X, A)$, the encoder compresses the graph into a hidden matrix, and the decoder reconstructs the graph. For example, Variational Graph Auto-Encoder (VGAE) (Kipf & Welling, 2016b) embeds the input graph into the embedding matrix $X_{GAE}$, and reconstructs the graph adjacent matrix $\hat{A}$. The process is defined as $X_{VGAE} = f(X, A)$, $\hat{A} = g(X_{VGAE})$, where $f(\cdot)$ is the encoder function, and $g(\cdot)$ is the decoder function. With other strategies, GAEs have multiple improving methods (Hou et al., 2022; Tan et al., 2023; Li et al., 2024b; Zhou et al., 2025; Bai et al., 2024). For example, Graph-MAE (Hou et al., 2022) adopts the masking approach in the encoding process, and re-masks in the decoding process.

**Graph Contrastive Learning (GCL).** Distinguished from the GAEs, GCL methods generate the graph representations by comparing the positive samples against the negative ones (Zhu et al., 2021a). Therefore, the core of the GCL methods is how to obtain the adaptive samples, and the improvement is based on the data augmentation for sam-

pling (You et al., 2020). For instance, Deep Graph Infomax (DGI) (Veličković et al., 2019), a classical GCL method, adopts the corruption function to sample the negative graph.

# 3. Methodology

We propose SFCLTA, a novel GCL model to address the challenges mentioned in Section 1, which include the distribution shift and the loss of robustness. The overview of the proposed SFCLTA is shown in Figure 1. The proposed framework is composed of two parts: the topology-adaptive graph augmentation module, and the spectral fusion contrastive learning module. Combined with the corresponding loss calculations, these modules enable the SFCLTA to effectively learn the representations from the heterophilic graphs.

## 3.1. Topology-Adaptive Graph Augmentation

Preserving high-frequency signals is paramount in GCL for heterophilic graphs, as it captures the local dissimilarities essential for accurate representation learning. However, traditional data augmentation techniques operate agnostically to this spectral property. Random structural perturbations inevitably corrupt these high-frequency components, inducing the distribution shift that undermines the contrastive learning objective. To address this challenge in data augmentation, we propose the TAGA that generates spectrally stable views. The idea of our augmentation method is to adaptively rewire the graph topology based on the node feature similarity and the global heterophily, rather than applying fixed or random perturbations.

Given the input graph $G = (V, E)$ where $V$ is the node set with $n$ nodes and $E$ is the edge set without self-loops, we introduce our sampling strategy in Algorithm 1 to select the edges for topological change, which generates the augmented graph $G'$ with a new adjacent matrix $A'$. Notice that, the input graphs are usually large and sparse where $n$ may be over 10000, and the edge number $|E| \ll O(n^2)$ as shown in Table 4. The topology-aware data augmentation cannot evaluate all possible edge modifications in the complete graph. Therefore, the topological change process carefully considers edge sampling strategies.

The proposed algorithm adopts a learning-based method to calculate the possibilities of each potential edge. To realize the automatic learning of the $w_{i,j}$, we design the corresponding loss function for training. In detail, the global homophily metric is computed as below

$$H = \frac{1}{|E|} \sum_{\forall (v_o, v_s) \in E} \cos(x_o, x_s), \qquad (2)$$

where $\cos(\cdot)$ is the cosine similarity between the feature vectors. The augmentation result is a new graph with different heterophily from the original graph, and the loss calculation

---

**Algorithm 1** The sampling algorithm for data augmentation.

**Input:** graph $G = (V, E)$ with feature matrix $X$ and adjacent matrix $A$, data augmentation ratio $r_{da} \in (0, 1)$.
**Output:** augmented graph $G'$.
1: Number of changing edges $n_{add} \leftarrow \lfloor r_{da} \cdot n \rfloor$.
2: Candidate node set $V_{can} \leftarrow V$.
3: Initialize $A' \leftarrow A$.
4: Select node $v_i$ whose degree is the highest in $V_{can}$.
5: Initialize $V'_{can} \subset V_{can} - \{v_i\}$, where $|V'_{can}| = n_{add}$.
6: $\mathcal{V} \leftarrow \{(v_i, v_j)\}$, where $\forall v_j \in V'_{can}$.
7: **for** each $(v_i, v_j) \in \mathcal{V}$ **do**
8:     $w_{i,j} \leftarrow \sigma(x_i, x_j)$, where $\forall x_i, x_j \in X$ are the feature vectors of the pair $(v_i, v_j)$, and $\sigma(\cdot)$ is the MLP with sigmoid function.
9:     Update $A'_{i,j} \leftarrow w_{i,j}, A'_{j,i} \leftarrow w_{i,j}$.
10: **end for**
11: Update graph edge set to $E'$ according to the adjacent matrix $A'$.
12: $G' \leftarrow (V, E')$.

---

further enlarges the heterophily distance between the augmented graph and the original graph. Therefore, the loss function $\mathcal{L}_1$ is improved as follows.

$$\mathcal{L}_1 = -||H_{\mathrm{IG}} - H_{\mathrm{AG}}||_2 + \mathrm{tr}((A - A')^\top L(A - A')), \quad (3)$$

where $H_{\mathrm{IG}}$ is the global homophily value of the input graph, $H_{\mathrm{AG}}$ is the global homophily value of the augmented graph, and $\mathrm{tr}(\cdot)$ is the trace of the matrix. Since the goal of the graph augmentation is to obtain a graph with different heterophily, we adopt the difference between $H_{IG}$ and $H_{AG}$. The second term $\mathrm{tr}((A - A')^\top L(A - A'))$ explicitly constrains high-frequency distortion by Dirichlet energy principle (Zhou et al., 2021).

## 3.2. Spectral Fusion Contrastive Learning

With the augmented graph with different heterophily, we naturally adopt the GCL framework which leverages positive and negative samples. To extract the multi-frequency signals, the encoder is composed of two modules, including the LPF and the HPF. Given the input original graph $G = (X, A)$ and the augmented graph $G' = (X, A')$, we separately define the LPF and the HPF operators for the corresponding views. The encoder modules of the input graph are defined as follows

$$F_{\mathrm{L}} = \mathrm{ReLU}(\tilde{A}XW_{\mathrm{lpf}}), \; F_{\mathrm{H}} = \mathrm{ReLU}(\tilde{L}XW_{\mathrm{hpf}}), \quad (4)$$

where $F_{\mathrm{L}}, F_{\mathrm{H}} \in \mathbb{R}^{n \times d_h}$ with the hidden dimension $d_h$ are respectively the low-pass feature matrix and the high-pass feature matrix, $W_{\mathrm{lpf}}, W_{\mathrm{hpf}} \in \mathbb{R}^{d \times d_h}$ are respectively the LPF's learnable weight matrix and the HPF's learnable weight matrix, $\tilde{L} = \tilde{D} - \tilde{A}$ is the Laplacian matrix with the

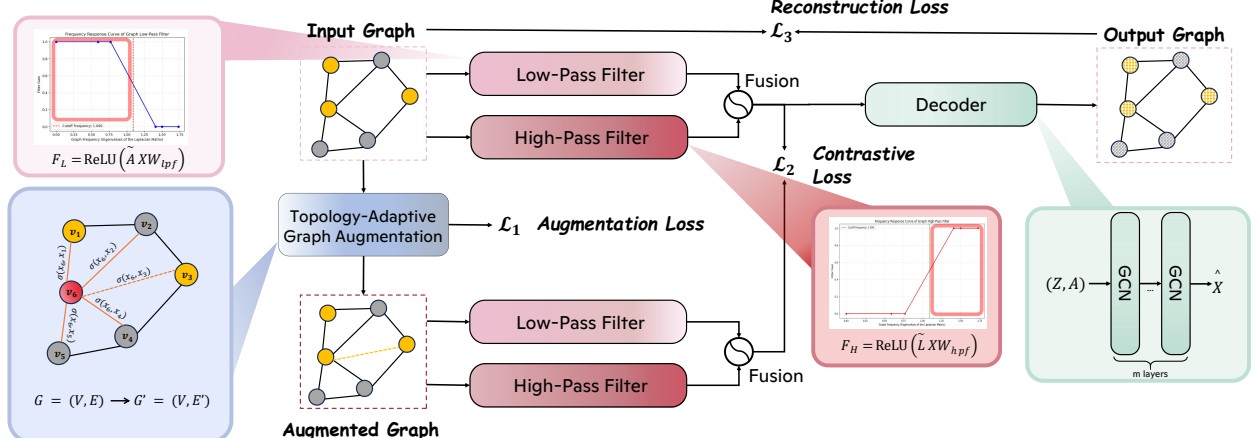

*Figure 1.* The architecture of our proposed SFCLTA. This framework for training is composed of three parts: topology-adaptive graph augmentation (blue), spectral fusion contrastive learning (red & pink), and feature reconstruction (teal). In the TAGA module, the data augmentation process updates the edge weights by MLP (i.e., $\sigma(\cdot)$). In the SFCL, we highlight the low- and high-frequency signals in a graph, and separately show the Frequency Response Curves in the upper part. With the corresponding modules and loss functions, the proposed model learns the multi-frequency fusion representations.

self-loops, and $\text{ReLU}(\cdot)$ is the ReLU activation function. The encoder modules of the augmented graph are defined as

$$F'_{\text{L}} = \text{ReLU}(\tilde{A}'XW'_{\text{lpf}}), \ F'_{\text{H}} = \text{ReLU}(\tilde{L}'XW'_{\text{hpf}}), \quad (5)$$

where $F'_{\text{L}}, F'_{\text{H}} \in \mathbb{R}^{n \times d_h}$ are respectively the low-pass feature matrix and the high-pass feature matrix for the augmented graph, and the $W'_{\text{lpf}}, W'_{\text{hpf}} \in \mathbb{R}^{d \times d_h}$ are the corresponding weights of the augmented graph, and the Laplacian matrix $\tilde{L}' = \tilde{D}' - \tilde{A}'$. Note that, the $W_{\text{lpf}}$ shares the parameters with the $W'_{\text{lpf}}$, and similarly the $W_{\text{hpf}}$ shares parameters with the $W'_{\text{hpf}}$. The sample pairs (the original graph & the augmented graph) should maintain semantic consistency within the same feature space.

With the low- and high-frequency features from the two view graphs, we aim to fuse the multi-frequency features belonging to the same graph for representation learning. Therefore, we adopt the weighted high-frequency strategy to mitigate noise sensitivity, reserving the discriminative high-frequency signal. The weighting process is realized by

$$\tilde{F}_{\text{H}} = W_{\text{H}} \odot F_{\text{H}}, \ \tilde{F}'_{\text{H}} = W'_{\text{H}} \odot F'_{\text{H}}, \quad (6)$$

where $W_{\text{H}}, W'_{\text{H}} \in \mathbb{R}^{n \times d_h}$ are the weight matrices, and $\odot$ is the Hadamard product operation. Then, we fuse the multi-frequency features as follows

$$Z = \text{Concat}(F_{\text{L}}, \tilde{F}_{\text{H}}), \ Z' = \text{Concat}(F'_{\text{L}}, \tilde{F}'_{\text{H}}), \quad (7)$$

where $\text{Concat}(\cdot)$ is the concatenation operation, and $Z, Z' \in \mathbb{R}^{n \times 2d_h}$.

**Output Representation.** The fusion matrix $Z$ is the output representation for the downstream tasks. For the graph-level

task, we utilize the READOUT function to aggregate node representations into a graph embedding.

Subsequently, with these fused representations, the contrastive learning loss is formalized by the equation below.

$$\mathcal{L}_2 = -\sum_{\forall p \in V} \log \frac{\exp(\cos(Z_p, Z'_p)/\tau)}{\sum_{\substack{\forall q \in V^q_{k-\text{hop}}, \\ q \neq p}} \exp(\cos(Z_p, Z_q)/\tau)}, \quad (8)$$

where $V^q_{k-\text{hop}}$ is the node set of the $k$-hop subtree whose root node is $q$, and $\tau$ is a temperature parameter. Since the negative sampling is restricted to the local neighborhoods, we reduce the time complexity of $\mathcal{L}_2$ to $O(kn)$ where $O(kn) \ll O(n^2)$.

### 3.3. Feature Reconstruction

With the aforementioned modules, we obtain the multi-frequency fusion representations. The topology-aware augmentation alleviates the distribution shift problem. Contrastive learning alone may fail to preserve sufficient feature information aligned with the original graph, resulting in a lack of semantics. So we adopt the feature reconstruction task as the pretext task to enhance the representations.

Given the output matrix $Z \in \mathbb{R}^{n \times 2d_h}$ from the input graph, the pretext task aims to reconstruct features from $Z$. We adopt the multi-layer GCN module as the decoder. The reconstruction process is realized as below

$$\hat{X} = \underset{m-\text{layers}}{\text{GCN}} (Z, A), \quad (9)$$

where $\hat{X}$ is the reconstruction representation, $\underset{m-\text{layers}}{\text{GCN}}$ is the multi-layer GCN with $m$ layers. We calculate the similarity

between the reconstruction features and the original features with the following equation

$$\mathcal{L}_3 = \frac{1}{n}||\hat{X} - X||_2. \tag{10}$$

With the Equation 3, 8 and 10, we respectively introduce the corresponding loss function for each module. We utilize the global loss $\mathcal{L} = \mathcal{L}_1 + \mathcal{L}_2 + \mathcal{L}_3$ to train the proposed model.

### 3.4. Theoretical Analysis

Next, to validate the effectiveness of our method in theory, we prove that the proposed SFCLTA achieves the control of the distribution shift and the enhancement of the HPF's robustness. We encourage readers to read the Appendix first for a detailed discussion on the challenges and the theorems.

**Theorem 1.** *G and $G'$ are the original and augmented graphs. L and $L'$ are the Laplacian matrices of G and $G'$. The augmentation process is guided by the proposed topology-adaptive mechanism with spectral regularization loss function $\mathcal{L}_1$.*

*For the graph signals, there is a constant $C > 0$ such that*

$$D_{\mathrm{KL}}\left(P_{\mathrm{HF}}^G \| P_{\mathrm{HF}}^{G'}\right) \le C \cdot \|L - L'\|_2, \tag{11}$$

*where $D_{KL}(\cdot)$ is KL-divergency function, $P_{HF}^G$, $P_{HF}^{G'}$ are the high-frequency spectral energy distribution of G and $G'$.*

This theorem shows that the augmented graph $G'$ maintains spectral consistency with $G$. The high-frequency distributional shift is strictly bounded. Compared to the SFCLTA, traditional methods do not provide the upper-bound control, resulting in unlimited or even tremendous distribution shifts in the data augmentation. The augmented graph with flawed and noisy high-frequency signals cannot provide effective information for the GCL. The spectral constraint $\mathcal{L}_1$ utilized in the SFCLTA guarantees that augmentation will not degrade informative high-frequency signals. Please refer to Appendix A.4 for more proofs.

**Theorem 2.** *If the high-pass filtered node-level representation is defined as $\mathbf{z} = L\mathbf{x} + L\boldsymbol{\epsilon}$ where $\boldsymbol{\epsilon}$ is the additive Gaussian noise, the node confidence could be defined as $\alpha = \exp(-\mathcal{H}) \in (0,1]$ where $\mathcal{H}$ is the entropy of the high-frequency components of node. Then, the expected deviation between the ideal and noisy contrastive gradients satisfies*

$$\mathbb{E}_{\boldsymbol{\epsilon}}(||\nabla\mathcal{L}_2 - \nabla\mathcal{L}_{\mathrm{ideal}}||_2) \le (1 - \alpha_{\min}) \cdot C_2 \cdot \xi, \tag{12}$$

*where $\mathcal{L}_{\mathrm{ideal}}$ is the ideal loss value without noise, $\alpha_{\min}$ is the minimum value in node confidences, $C_2$ is a constant related to the temperature $\tau$, and $\xi$ is the standard deviation of the noise.*

Based on Theorem 2, we could summarize the robustness advantage of the proposed SFCLTA in handling noisy high-pass signals. Unlike the traditional GCL framework, the

SFCLTA introduces frequency-aware confidence weighting, which adaptively down-weights the contribution of the noisy nodes, thereby suppressing instability. Moreover, by leveraging the $k$-hop negative sampling, SFCLTA focuses on representation alignment within the local structures. This operation mitigates the risk of the contrastive collapse due to long-range graph noise, which is especially problematic in the heterophilic settings.

**Theorem 3.** *Given the original node feature $\mathbf{x}$ and the reconstructed node feature $\hat{\mathbf{x}}$, the loss function $\mathcal{L}_3$ implies to minimize $\mathcal{L}_3^{node} = ||\mathbf{x} - \hat{\mathbf{x}}||_2$, that is, the high-frequency spectral distributions of $\mathbf{x}$ and $\hat{\mathbf{x}}$ are aligned.*

$$D_{\mathrm{KL}}\left(P_{\mathrm{HF}}^{\mathbf{x}} \| P_{\mathrm{HF}}^{\hat{\mathbf{x}}}\right) \le \varepsilon, \text{where } \varepsilon \to 0 \text{ as } \mathcal{L}_3^{\mathrm{node}} \to 0 \tag{13}$$

Theorem 3 highlights the benefit of the feature reconstruction task. Although the contrastive learning modules align representations implicitly, the reconstruction task provides an explicit alignment in both spatial and frequency domains. As a result, the learned embeddings maintain more consistent semantics under augmentation and perturbation.

### 3.5. Discussions of Multi-Frequency Signals in Graphs

Compared to the GCL with multi-frequency signals, our proposed SFCLTA has several important properties, which interpret the effectiveness. First, unlike the traditional frequency-aware GCL methods (Chen et al., 2024; Yang & Mirza-soleiman, 2024), our approach focuses on contrastive learning on fused signals, not separate contrastive learning on different frequency signal views. Thus, our proposed method could achieve the complementarity of high-frequency and low-frequency signals in heterophily graphs, i.e., the effective representations of heterophily graphs require both the global features from the LPF and the local features from the HPF. So the fusion operation can significantly improve the representations for heterophily graphs. Second, compared to the popular spectral fusion methods in other applications (e.g., hyperspectral image tasks), our proposed SFCLTA improves the high-frequency signals in the fusion operation. In Section 2, we clearly point out that graphs are non-Euclidean data, distinguished from Euclidean data such as images. Traditional spectral fusion methods (e.g., matrix decomposition-based fusion) cannot be directly applied to the topological dependence of graphs. We re-design the spectral fusion and adopt its module in GCL. Specifically, we first integrate a confidence-weighted refinement strategy to prioritize discriminative high-frequency components while reducing noisy ones, then perform a synergistic fusion of these optimized high-frequency features with globally informative low-frequency signals. This two-step process ensures that the fused representations retain the complementary strengths of both frequency domains, rather than simply combining raw signal components.

# 4. Experiments

In this section, we introduce the experiments and evaluate the proposed model by answering the following questions.

- **RQ1.** Does the proposed model outperform the existing baseline models on node- and graph-level tasks?

- **RQ2.** Does the proposed graph augmentation approach benefit the representation learning of SFCLTA? How does LPF or HPF affect model performance?

- **RQ3.** Is the proposed model robust to the noise?

- **RQ4.** Is our model sensitive to hyper-parameters?

We show a brief description of datasets, experimental settings and baselines in Appendix B. Then, we proceed to the details of experimental results and their analysis.

## 4.1. Results (RQ1)

The results of the node classification task and the graph classification task are presented in Table 1 and Table 2. From these results, we clearly observe that our proposed model, SFCLTA, achieves outstanding performance not only in the node classification task but also in the graph classification task. We analyze the results as follows.

**First**, the graphs with high heterophily are the main challenges for the traditional GNN models, such as GCN (Kipf & Welling, 2016a), VGAE (Kipf & Welling, 2016b), relying on the homophily principle. Models designed for heterophily graphs like H2GCN (Zhu et al., 2020) significantly outperform traditional GNNs on heterophilic graphs, highlighting the importance of breaking the homophily assumption. The proposed SFCLTA further outperforms these traditional GNNs on all datasets, demonstrating its strong generalization ability on the graphs with different heterophily values. Due to the heterophily-aware design in the graph augmentation process, the SFCLTA is applicable to various graphs.

**Second**, self-supervised learning, especially GCL, is effective for improving graph representations, with typical methods (e.g., DGI (Veličković et al., 2019), PolyGCL (Chen et al., 2024), HLCL (Yang & Mirzasoleiman, 2024)) showing competitiveness. Among them, frequency-based approaches like PolyGCL and HLCL perform remarkably, verifying the necessity of multi-frequency signals. However, these methods lack mechanisms to guarantee high-frequency distribution and have limitations in robustness and stability. SFCLTA addresses these via topology-adaptive augmentation (alleviating high-frequency shift), enhanced HPF noise resistance, and a feature reconstruction task (preserving feature semantics). Results in Table 1 confirm its superior representation learning over other GCL baselines.

**Third**, although SFCLTA is primarily designed for the node-level representation learning, it also demonstrates strong performance on the graph-level tasks. From Table 2, the outperformance of our proposed model highlights the generalization ability of the representations and confirms the effectiveness of multi-frequency fusion not only for capturing local node semantics but also for capturing global graph-level information.

## 4.2. Ablation Study (RQ2)

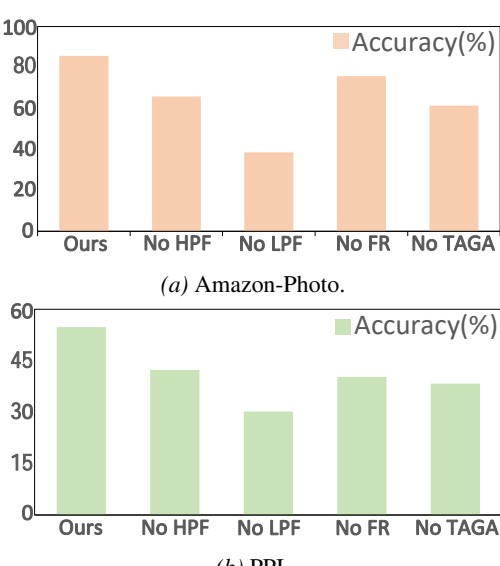

*(a)* Amazon-Photo.

*(b)* PPI.

*Figure 2.* Ablation experiments on Amazon-Photo and PPI.

To explore the effectiveness of each module in the proposed model, we set the ablation experiments on Amazon-Photo and PPI. The results of these experiments are shown in Figure 2. Specifically, **Ours** is the proposed SFCLTA, **No-HPF** is the one without the HPF modules in double views; **No-LPF** is the one without the LPF modules in double views; **No-FR** is the one without the pretext task feature reconstruction; **No-TAGA** is the one which replaces the TAGA module with random edge rewiring. Note that the contrastive learning paradigm in No-TAGA is not changed.

Our observations are as below. Removing HPF or LPF from SFCLTA reduces performance, showing both frequency components are essential and complementary, enabling generalization across graphs with different homophily. No-FR without feature reconstruction relies only on contrastive learning, suffering from semantic distortion and performance drop, proving contrastive learning alone is insufficient, and feature reconstruction is necessary. No-TAGA adopts fixed pre-trained graph augmentation, which fails to mitigate distribution shift during training and thus underperforms SFCLTA. This underscores the criticality of the problem and the efficacy of our method.

*Table 1.* Node classification accuracy (%) with the standard error on the datasets.

| Method | Cora | CiteSeer | Amazon-Computers | Amazon-Photo | PPI | Roman-Empire | Amazon-Ratings | Squirrel |
|---|---|---|---|---|---|---|---|---|
| GCN | 80.1 ± 0.2 | 70.3 ± 0.1 | 73.0 ± 0.4 | 78.4 ± 0.3 | 42.0 ± 1.2 | 73.5 ± 0.9 | 41.2 ± 0.8 | 29.8 ± 0.7 |
| VGAE | 78.5 ± 0.6 | 69.0 ± 0.4 | 71.5 ± 0.5 | 80.2 ± 0.5 | 45.5 ± 1.8 | 76.2 ± 0.8 | 42.3 ± 0.9 | 30.1 ± 0.6 |
| DGI | 81.9 ± 0.1 | 71.8 ± 0.2 | 75.5 ± 0.5 | 83.5 ± 0.3 | 54.7 ± 1.1 | 78.2 ± 0.7 | 45.1 ± 0.6 | 33.5 ± 0.8 |
| H2GCN | 82.3 ± 0.2 | 72.4 ± 0.3 | 78.2 ± 0.3 | 84.1 ± 0.7 | 53.2 ± 1.0 | 60.7 ± 0.8 | 40.1 ± 0.5 | 46.5 ± 0.6 |
| GraphMAE | 83.0 ± 0.1 | 72.8 ± 0.2 | 79.1 ± 0.2 | 84.8 ± 0.4 | 50.9 ± 1.1 | 81.2 ± 0.7 | 51.8 ± 0.4 | 42.0 ± 0.5 |
| PolyGCL | 83.1 ± 0.2 | 73.0 ± 0.2 | 79.8 ± 0.3 | 85.0 ± 0.9 | 54.1 ± 1.4 | 84.9 ± 0.6 | 52.3 ± 0.5 | **47.5 ± 0.5** |
| HLCL | 83.3 ± 0.3 | 73.3 ± 0.2 | 80.2 ± 0.3 | 85.2 ± 0.6 | 52.4 ± 1.1 | 84.5 ± 0.6 | 52.7 ± 0.4 | 44.9 ± 0.4 |
| **SFCLTA(Ours)** | **84.1 ± 0.2** | **74.2 ± 0.2** | **82.8 ± 0.6** | **86.1 ± 0.9** | **54.8 ± 1.7** | **85.3 ± 0.5** | **54.3 ± 0.3** | 46.7 ± 0.3 |

*Table 2.* Graph classification accuracy (%) with the standard error on the datasets.

| Method | PROTEINS | NCI1 | IMDB-B | IMDB-M |
|---|---|---|---|---|
| GCN | 75.3 ± 0.5 | 76.4 ± 0.4 | 72.6 ± 0.6 | 49.5 ± 0.5 |
| DiffPool | 76.2 ± 0.6 | 77.8 ± 0.5 | 73.8 ± 0.5 | 50.3 ± 0.4 |
| VGAE | 73.1 ± 0.9 | 75.1 ± 0.6 | 71.8 ± 0.6 | 48.2 ± 0.5 |
| DGI | 75.7 ± 0.5 | 78.1 ± 0.8 | 73.2 ± 0.6 | 50.1 ± 0.5 |
| GraphMAE | 75.5 ± 0.4 | 79.0 ± 0.4 | 74.1 ± 0.7 | 51.3 ± 0.4 |
| QSGCNN | 75.9 ± 0.8 | 77.5 ± 0.9 | 73.6 ± 1.1 | 51.6 ± 1.2 |
| AEGK | 75.1 ± 0.3 | 75.5 ± 0.6 | 73.5 ± 0.5 | 50.3 ± 0.8 |
| HLCL | 77.9 ± 0.4 | 79.2 ± 0.3 | 74.3 ± 0.4 | 51.8 ± 0.3 |
| **SFCLTA(Ours)** | **78.5 ± 0.3** | **80.1 ± 0.3** | **74.9 ± 0.4** | **52.4 ± 0.3** |

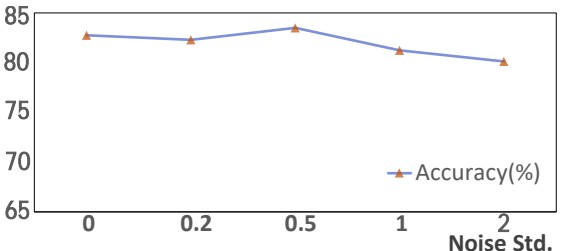

*(a)* Noise Added in HPF.

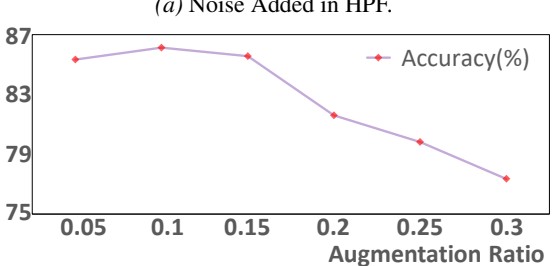

*(b)* Augmentation Ratio.

*Figure 3.* Experiments on Amazon-Photo. Figure (a) shows the results of the SFCLTA with the Gaussian noise. Figure (b) shows the results of the SFCLTA with different augmentation ratios.

## 4.3. Sensitivity Analysis (RQ3 & RQ4)

### 4.3.1. NOISE ADDED IN HPF (RQ3)

Additionally, we set sensitive experiments to evaluate the robustness of the HPF. The results are shown in Figure 3a, where Noise Std. denotes the standard deviation of the Gaussian noise. Even though our model has Gaussian noise in training, its performance is stable. It verifies that the robustness of the HPF is strong enough so that the influence of noise is negligible.

### 4.3.2. AUGMENTATION RATIO (RQ4)

In the experiments on the Amazon-Photo dataset, we set the augmentation ratio $r_{da}$ from the set {0.05, 0.1, 0.15, 0.2, 0.25, 0.3}. The results are shown in Figure 3b. We observe that when $r_{da}$ is set in {0.05, 0.1, 0.15}, the accuracy is stable (i.e., over 85%). $r_{da} = 0.1$ is the best ratio among the candidate set. However, when $r_{da}$ keeps increasing continuously and exceeds 0.2, the performance of the proposed model obviously drops. The growth of the augmentation ratio $r_{da}$ implies an increase in the number of potential edges. With the fixed training epoch, too much complex input data may exceed the learning capacity of the naive neural network (Burbidge et al., 2001). We need to carefully select the augmentation ratio $r_{da}$.

## 5. Conclusion

In this paper, we address the challenges of distribution shift and loss of robustness in unsupervised GCL, especially under heterophilic graph settings. We propose SFCLTA that in-tegrates topology-adaptive graph augmentation with spectral fusion contrastive learning. By dynamically adjusting graph topology and applying spectral regularization, SFCLTA effectively controls high-frequency perturbations. Furthermore, the confidence-weighted fusion of multi-frequency signals enhances the robustness of highpass filtering, and the feature reconstruction task explicitly mitigates distribution shifts at the feature level. Experimental results validate the superior performance of SFCLTA on both node and graph classification tasks, especially on graphs with low homophily. In the future, we will explore the use of the proposed model in dynamic graphs and multi-view graphs.

## Acknowledgements

This work is supported by National Natural Science Foundation of China (No. 62576371, 62576198, and 62536006), and the Open Project Foundation of Key Laboratory of Computation Intelligence and Chinese Information Processing of Ministry of Education and Key Laboratory of Data Intelligence and Cognitive Computing of Shanxi Province.

## Impact Statement

This paper presents work whose goal is to advance the field of machine learning. There are many potential societal consequences of our work, none of which we feel must be specifically highlighted here.

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

# A. Proofs

## A.1. Explanation of Symbols in Proofs

We list the symbols utilized in the Proofs with their meanings in Table 3.

*Table 3.* Notation Summary for SFCLTA

| Symbol | Meaning |
| --- | --- |
| $G = (V, E)$ | Original graph with nodes and edges |
| $L, L'$ | Laplacian matrices of original, augmented graph |
| $U, \Lambda$ | Eigenvectors and eigenvalues of $\mathbf{L}$ |
| $\lambda_i$ | $i$-th eigenvalue of the Laplacian |
| $\mathbf{u}_i$ | $i$-th eigenvector of the Laplacian |
| $\lambda_c$ | Frequency cutoff to separate high- and low-frequency components |
| $\hat{\mathbf{x}}$ | Spectral projection $U^\top \mathbf{x}$ |
| $\hat{\mathbf{x}}_{\mathrm{HF}}$ | High-frequency projection of $\mathbf{x}$ |
| $\hat{\mathbf{x}}'$ | Projection under the augmented graph basis $U'$ |
| $P_{\mathrm{HF}}, Q_{\mathrm{HF}}$ | High-frequency energy distributions (before/after augmentation) |
| $D_{\mathrm{KL}}(P\|Q)$ | KL divergence between distributions $P$ and $Q$ |
| $\Delta \mathbf{u}$ | Perturbation of eigenvectors under topology change |
| $\delta$ | Minimum eigenvalue gap $\min_{i \neq j} |\lambda_i - \lambda_j|$ |
| $\mathcal{L}_1$ | Spectral alignment loss |
| $\mathcal{L}_2$ | Confidence-weighted InfoNCE loss |
| $\mathcal{L}_3$ | Feature reconstruction loss |
| $\mathbf{z}_i$ | High-pass filtered representation of node $i$ |
| $\epsilon_i$ | Gaussian noise injected into input feature |
| $\xi^2$ | Variance of injected noise |
| $\alpha_i$ | Confidence weight for node $i$ based on spectral entropy |
| $\mathcal{H}_i$ | High-frequency entropy of node $i$ |
| $\tau$ | Temperature parameter in InfoNCE loss |
| $\mathcal{N}_i$ | Neighbor node set of anchor node $i$ |
| $h_i$ | Node feature of node $i$ |
| $\bar{h}_i$ | Aggregated neighborhood feature of node $i$ |
| $H$ | Neighborhood label consistency score |
| $n$ | Number of neighbors or total nodes (context-dependent) |
| $\mathbb{E}_{\mathrm{HF}}(\mathbf{x})$ | High-frequency energy of signal $\mathbf{x}$ |
| $g(\lambda)$ | Low-pass filter function |
| $h_{\mathrm{HP}}(\lambda)$ | High-pass filter $1 - g(\lambda)$ |

## A.2. Distribution Shift

**A simple case.** We set the node $i$ as the anchor node. The feature of the node $i$ in GNN could be representaed as $\bar{h}_i = \mathrm{AGG}(\{h_j | \forall h_j \in \mathcal{N}_i\})$, where $\mathcal{N}_i$ is the neighbor node set including node $i$, and $\mathrm{AGG}(\cdot)$ is the aggregation operation. Through the aggregation, the feature of node $i$ is calculated as below

$$\bar{h}_i = \frac{1}{|\mathcal{N}_i|} \sum_{j}^{|\mathcal{N}_i|} h_j.$$

The relevance between node $i$ and its neighbor nodes is computed as follows

$$H = \frac{1}{|\mathcal{N}_i|} \sum_{j}^{|\mathcal{N}_i|} 1(y_i = y_j).$$

If the neighbor nodes and the anchor node **do not** follow the same feature distribution (e.g., the gaussian distribution), we assume that $X_i \sim \mathcal{N}(\mu_1, \sigma_1)$, $X_{neighor} \sim \mathcal{N}(\mu_2, \sigma_2)$, and the neighbor number is $n$. Through the aggregation process, the expectation and the variance of the anchor node is calculated as below

$$\mathrm{E}(\bar{X}_i) = \frac{\mu_1 + n\mu_2}{n+1}, \ \mathrm{Var}(\bar{X}_i) = \frac{\sigma_1^2 + n^2\sigma_2^2}{(n+1)^2}.$$

We can notice that the expectation and the variance of the node $i$ are changed when the neighbor number $n$ is large. There is an obvious distribution shift, where the anchor node hardly preserve the original distribution attributions. If the original graph follows the Scale-Free Network where a few nodes have lots of connections to other nodes, and lots of nodes only have a few connections. The anchor node after aggregation tend to follow $\mathcal{N}(\mu_2, \sigma_2)$, resulting in the performance drop on the node classification.

**Effect of topological change.** Given an input graph with Laplacian matrix $L = U\Lambda U^\top$, we assume that the changement of $L$ is $\Delta L$, and calculate the new Laplacian matrix $L'$ as below

$$L' = L + \Delta L,\ \Lambda' = \Lambda + \Delta\Lambda,\ U' = U + \Delta U.$$

The eigenvector stability is evaluated by the Davis-Kahan Theorem shown below

$$\|\Delta U\|_F \leq \frac{\|\Delta L\|_F}{\delta},$$

where $\delta = \min_{i \neq j} |\lambda_i - \lambda_j|$ is eigenvalue gap.

**Lemma 1.** The eigenvalue gap of the high-frequency signals is lower than the eigenvalue gap of the low-frequency signals.

*Proof.* We assume that the graph follows the Power Law Distribution. Eigenvalue density $\rho(\lambda)$ follows

$$\rho(\lambda) \propto \lambda^{-\alpha},\ \alpha \in (0, 1),$$

The Low-Frequency Signals ($\lambda \to 0^+$): $\rho(\lambda) \propto \lambda^{-\alpha} \to 0$ (due to $\alpha > 0$), So

$$\delta_{\text{low}} \approx \frac{1}{n\rho(\lambda)} \to \infty$$

The High-Frequancy Signals ($\lambda \to \lambda_{\max}^-$): Given $\rho(\lambda_{\max}) = c_0 > 0$, we could obtain

$$\delta_{\text{high}} \approx \frac{1}{nc_0} = \mathcal{O}\left(\frac{1}{n}\right)$$

**Remark 1.** $\delta_{\text{high}} \ll \delta_{\text{low}}$ when $n$ is too large. Thus, the high-frequency eigenvector perturbation as below

$$\|\Delta \mathbf{u}_{\text{high}}\| \leq \frac{\|\Delta L\|}{\delta_{\text{high}}} = \mathcal{O}(n\|\Delta L\|),$$

and the low-frequency eigenvector perturbation is computed as

$$\|\Delta \mathbf{u}_{\text{low}}\| \leq \frac{\|\Delta L\|}{\delta_{\text{low}}} = \mathcal{O}(\|\Delta L\|).$$

Therefore, we can notice that $\|\Delta \mathbf{u}_{\text{high}}\| \propto 1/\delta_{\text{high}} \gg \|\Delta \mathbf{u}_{\text{low}}\|$, and conclude that the high-frequency graph singals is more sensitive to the graph topological changes, comparing to the low-frequency graph singals.

### A.3. HPF's Robustness

**Weak robustness of the HPF.** According to the Remark 1, the high-frequency signals are more sensitive to the graph topological changes, comparing to the low-frequency graph singals. We assume that the low-pass filter is defined as $h_{\text{LP}}(\lambda) = g(\lambda)$ where $g(\cdot)$ is a decreasing function, and the high-pass filter is defined as $h_{\text{HP}}(\lambda) = 1 - g(\lambda)$. Given the Laplacian matrix $L = U\Lambda U^\top$, the high-pass filter could be expanded as

$$H_{\text{HP}} = I - g(L) = U(I - g(\Lambda))U^\top.$$

Given the noise $\eta \sim \mathcal{N}(0, \sigma^2 I)$, the output noise of the high-pass filter is

$$\Delta y_{\text{HP}} = U(I - g(\Lambda))U^\top \eta.$$

And the output noise energy is

$$\mathbb{E}[\|\Delta y_{\mathrm{HP}}\|_2^2] = \sum_{k=1}^n |1 - g(\lambda_k)|^2 \cdot \sigma^2.$$

When $\lambda_k \to \lambda_{\max}$, $|1 - g(\lambda_k)| \to 1$. And thus,

$$\mathbb{E}[\|\Delta y_{\mathrm{HP}}\|_2^2] \geq \sigma^2 \cdot |\{k : \lambda_k > \bar{\lambda}\}| = \Omega(n\sigma^2)$$

**Remark 2.** The noise is enlarged by the HPF. Therefore, the robustness of the HPF is weak.

### A.4. Theorem 1

Let $L = U\Lambda U^\top$ and $L' = U'\Lambda'U'^\top$ be the eigendecompositions of the original and augmented Laplacians. Let $\hat{\mathbf{x}} = U^\top \mathbf{x}$, $\hat{\mathbf{x}}' = U'^\top \mathbf{x}$ be the spectral projections.

We define the high-frequency index set $\mathcal{I}_{\mathrm{HF}} = \{i : \lambda_i > \lambda_c\}$. Then,

$$\mathbb{E}_{\mathrm{HF}}(\mathbf{x}) = \sum_{i \in \mathcal{I}_{\mathrm{HF}}} (\mathbf{u}_i^\top \mathbf{x})^2 = \|\hat{\mathbf{x}}_{\mathrm{HF}}\|^2,$$

$$\mathbb{E}_{\mathrm{HF}}(\mathbf{x}') = \sum_{i \in \mathcal{I}_{\mathrm{HF}}} (\mathbf{u}_i'^\top \mathbf{x})^2 = \|\hat{\mathbf{x}}_{\mathrm{HF}}'\|^2.$$

We aim to bound

$$|\mathbb{E}_{\mathrm{HF}}(\mathbf{x}') - \mathbb{E}_{\mathrm{HF}}(\mathbf{x})| \leq 2\|\hat{\mathbf{x}}_{\mathrm{HF}}\| \cdot \|\hat{\mathbf{x}}_{\mathrm{HF}} - \hat{\mathbf{x}}_{\mathrm{HF}}'\|,$$

by Davis–Kahan theorem

$$\|\hat{\mathbf{x}}_{\mathrm{HF}}' - \hat{\mathbf{x}}_{\mathrm{HF}}\| \leq C \cdot \|L - L'\|_2.$$

The result is

$$|\mathbb{E}_{\mathrm{HF}}(\mathbf{x}') - \mathbb{E}_{\mathrm{HF}}(\mathbf{x})| \leq C_1 \cdot \|L - L'\|_2$$

We define normalized high-frequency distributions as follows

$$P_i = \frac{(\mathbf{u}_i^\top \mathbf{x})^2}{\mathbb{E}_{\mathrm{HF}}(\mathbf{x})}, \; Q_i = \frac{(\mathbf{u}_i'^\top \mathbf{x})^2}{\mathbb{E}_{\mathrm{HF}}(\mathbf{x}')}.$$

Based on the Pinsker's inequality, we can obtain the following condition

$$\begin{aligned} D_{\mathrm{KL}}(P_{\mathrm{HF}}\|Q_{\mathrm{HF}}) &\leq 2\|P - Q\|_1^2 \\ &\leq C_2 |\mathbb{E}_{\mathrm{HF}}(\mathbf{x}') - \mathbb{E}_{\mathrm{HF}}(\mathbf{x})| \\ &\leq C\|L - L'\|_2, \end{aligned}$$

where $C = C_1 \cdot C_2$.

### A.5. Theorem 2

Let the confidence-weighted contrastive loss be

$$\mathcal{L}_2 = -\sum_i \alpha_i \log \frac{\exp(\cos(\mathbf{z}_i, \mathbf{z}_i')/\tau)}{\sum_{j \in \mathcal{N}_i} \exp(\cos(\mathbf{z}_i, \mathbf{z}_j)/\tau)},$$

where $\mathbf{z}_i = L(\mathbf{x}_i + \boldsymbol{\epsilon}_i)$, and $\boldsymbol{\epsilon}_i \sim \mathcal{N}(0, \xi^2\mathbf{I})$ denotes high-frequency noise. Let $\mathbf{z}_i^* = L\mathbf{x}_i$ be the clean representation and $\alpha_i = \exp(-\mathcal{H}_i) \in (0, 1]$ be the confidence derived from entropy.

Since $\mathbf{z}_i = \mathbf{z}_i^* + L\boldsymbol{\epsilon}_i$, we observe that the representation shift induced by noise is

$$\Delta\mathbf{z}_i = L\boldsymbol{\epsilon}_i, \text{ with } \mathbb{E}[\|\Delta\mathbf{z}_i\|^2] = \xi^2 \cdot \mathrm{tr}(L^\top L) = \xi^2 \sum_k \lambda_k^2$$

where $\lambda_k$ are eigenvalues of the Laplacian filter $L$. This implies that high-frequency filters (with large $\lambda_k$) amplify the noise variance.

Given the InfoNCE loss structure, the gradient with respect to $\mathbf{z}_i$ is Lipschitz continuous. Let $g(\cdot) = \nabla_{\mathbf{z}_i}\mathcal{L}_2$, then

$$\|g(\mathbf{z}_i) - g(\mathbf{z}_i^*)\|_2 \leq L \cdot \|\mathbf{z}_i - \mathbf{z}_i^*\|_2 = L \cdot \|L\boldsymbol{\epsilon}_i\|_2$$

Taking the expectation over noise and summing over all nodes with confidence weights as

$$\mathbb{E}_{\boldsymbol{\epsilon}}\left[\|\nabla\mathcal{L}_2 - \nabla\mathcal{L}_2^*\|_2\right] \leq \sum_i \alpha_i \cdot \mathbb{E}\left[\|g(\mathbf{z}_i) - g(\mathbf{z}_i^*)\|_2\right]$$

$$\leq L\xi \sum_i \alpha_i \sqrt{\operatorname{tr}(L^\top L)}.$$

Since $\alpha_i \leq 1$ and $\alpha_i \to 0$ as $\mathcal{H}_i \to \infty$, we obtain

$$\mathbb{E}_{\boldsymbol{\epsilon}}\left[\|\nabla\mathcal{L}_2 - \nabla\mathcal{L}_2^*\|_2\right] \leq C \cdot \xi \cdot (1 - \alpha_{\min})$$

for some constant $C$ depending on the filter norm and temperature. This shows that the gradient deviation caused by high-frequency noise is suppressed by entropy-based confidence weights.

### A.6. Theorem 3

We consider the spectral decomposition of the input and reconstructed features. Let the Laplacian $\mathbf{L} = U\Lambda U^\top$, where $U = [\mathbf{u}_1, \ldots, \mathbf{u}_n]$ is the matrix of eigenvectors. Then we calculate

$$\mathbf{x} = \sum_{i=1}^n \alpha_i \mathbf{u}_i, \ \hat{\mathbf{x}} = \sum_{i=1}^n \hat{\alpha}_i \mathbf{u}_i.$$

The high-frequency component distribution is defined over the set $\{\lambda_i > \lambda_c\}$ as

$$P_{\text{HF}}^{\mathbf{x}}(i) = \frac{\alpha_i^2}{\sum_{\lambda_j > \lambda_c} \alpha_j^2}, \quad P_{\text{HF}}^{\hat{\mathbf{x}}}(i) = \frac{\hat{\alpha}_i^2}{\sum_{\lambda_j > \lambda_c} \hat{\alpha}_j^2}$$

From the assumption $\|\mathbf{x} - \hat{\mathbf{x}}\|^2 \to 0$, we have

$$\sum_{i=1}^n (\alpha_i - \hat{\alpha}_i)^2 \to 0 \ \Rightarrow \ \alpha_i^2 \approx \hat{\alpha}_i^2 \text{ for all } i.$$

Consequently, the KL divergence between the distributions satisfies

$$D_{\text{KL}}\left(P_{\text{HF}}^{\mathbf{x}} \| P_{\text{HF}}^{\hat{\mathbf{x}}}\right) = \sum_i P_{\text{HF}}^{\mathbf{x}}(i) \log\left(\frac{P_{\text{HF}}^{\mathbf{x}}(i)}{P_{\text{HF}}^{\hat{\mathbf{x}}}(i)}\right) \to 0$$

Hence, minimizing reconstruction loss in the node feature space guarantees convergence in the spectral high-frequency domain, ensuring frequency-preserving representation learning.

## B. Details of Experiments

### B.1. Datasets

We adopt eight real-world public benchmark datasets for the node classification task, and six datasets for the graph classification task. The details of these datasets are shown in the Table 4 and the Table 5. Note that, the datasets for the node classification task are labeled with the homophily metric $H_{\text{edge}}$, which implies we adopt abundant datasets with different

heterophily values. Note that, datasets such as Cora, CiteSeer, Amazon-Computers, Amazon-Photo have a low heterophily value ($H_{\text{edge}} \geq 0.5$), but the other datasets including PPI, Roman-Empire, Amazon-Ratings, Squirrel are high heterophilic. The calculation of $H_{\text{edge}}$ is defined as below

$$H_{\text{edge}} = \frac{|\{e_{u,v}|e_{u,v} \in E, y_u = y_v\}|}{|E|}, \tag{14}$$

where $E$ is the edge set, $e_{u,v}$ is the edge in $E$, and $y_u, y_v$ are respectively the node labels of the node $u$ and the node $v$.

Table 4. The datasets for the node classification task.

| Name | # Nodes | # Edges | # Features | # Classes | $H_{\text{edge}}$ |
|---|---|---|---|---|---|
| Cora | 2708 | 10556 | 1433 | 7 | 0.8100 |
| CiteSeer | 3327 | 9104 | 3703 | 6 | 0.7355 |
| Amazon-Computers | 13752 | 491722 | 767 | 10 | 0.7772 |
| Amazon-Photo | 7650 | 238162 | 745 | 8 | 0.8272 |
| PPI | 1767 | 32318 | 50 | 121 | 0.0007 |
| Roman-Empire | 22662 | 32927 | 300 | 18 | 0.0469 |
| Amazon-Ratings | 24492 | 93050 | 300 | 5 | 0.3804 |
| Squirrel | 5201 | 217073 | 2089 | 5 | 0.2234 |

Table 5. The datasets for the graph classification task.

| Name | # Nodes(Mean) | # Node Labels | # Graphs | #Classes |
|---|---|---|---|---|
| PROTEINS | 39.06 | 3 | 1113 | 2 |
| NCI1 | 29.87 | 37 | 4110 | 2 |
| IMDB-B | 19.77 | Null | 1000 | 2 |
| IMDB-M | 13.00 | Null | 1500 | 3 |

## B.2. Experimental Settings

To ensure the fairness of the our experiments, we compare our proposed model with the baseline models on the same experimental settings. We adopt the Adam optimizer which is set with the same learning rate of $1e-4$. Each model in the experiments is trained with 200 epochs. Moreover, the hidden dimension is set as 128, and the dropout rating is set as 0.5. We adopt the 10-fold cross validation to compute the classification accuracy. Following the previous work (Hou et al., 2022), we adopt the SVM to classify the output representations. All models in the experiments are trained with 4 NVIDIA GeForce RTX 3090s whose storage is 24 GB.

## B.3. Baselines

To ensure a fair comparison in the node classification task, we evaluate the proposed SFCLTA against four typical types of GNN-based baselines:

- Traditional GNN methods including GCN (Kipf & Welling, 2016a), VGAE (Kipf & Welling, 2016b).

- Graph self-supervised learning methods containing DGI (Veličković et al., 2019), GraphMAE (Hou et al., 2022). In detail, DGI adopt the GCL framework to maximize local mutual information to train the model, and GraphMAE based on the GAE framework learns representations by reconstructing node features.

- GNN model for heterophilic graphs such as H2GCN (Zhu et al., 2020), which could be adaptive in the graphs with different heterophily values.

- Frequency-aware GCL models comprising PolyGCL (Chen et al., 2024), HLCL (Yang & Mirzasoleiman, 2024), which are recent state-of-the-art (SOTA) approaches leveraging multi-frequency signals for heterophilic graphs.

In the graph classification task, we also evaluate multiple baselines, GCN (Kipf & Welling, 2016a), DiffPool (Ying et al., 2018), QSGCNN (Bai et al., 2023), and AEGK (Bai et al., 2025), which are widely utilized in supervised learning. Beyond these two baselines, the others are self-supervised methods.

