# OpenReview forum: "SFCLTA: Spectral Fusion Contrastive Learning with Topology-Adaptive Graph Augmentation"
_ICML.cc/2026/Conference — ICML 2026 regular_

### Official Review · Reviewer_WEnx · 2026-03-09

**Soundness:** 2
**Presentation:** 2
**Significance:** 2
**Originality:** 3
**Overall Recommendation:** 4
**Confidence:** 3

**Summary:**

This paper studies contrastive learning from the perspective of graph spectrum. Specifically, the authors foucs on retaining the high-frequency part during augmentation. For methodology, they propose TAGA augmentation that increase the homophily gap while constrain high-frequency distortion.

**Compliance With Llm Reviewing Policy:**

Affirmed.

**Final Justification:**

As the authors claimed, this paper "only limits how much the high-frequency distribution can be modified". That means they did not change high-frequency a lot across views. However, [1] proved that we need to make a larger modification to high-frequency part, so that the mode can cancel out the influence from it and so benifit homogeneous graphs. This is because the essence of contrastive learning is to keep unchanged information in two views. If we modify the high-frequency a lot, then low-frequency part is relatively unchanged, and the model can keep low-frequency part based on contrastive learning.

I can increase my score, because I am not sure if these two opinions are totally different, or can reach an agreement at a deeper level. For example, maybe we need to make larger augmentation in high-frequency part, but not too much? This could be an interesting topic. But finally, I stand by [1] that if we want to keep one part, we need to augment the other part.

**Key Questions For Authors:**

Please refer to the above weaknesses.

**Limitations:**

The authors did not discuss the limitations and potential negative societal impact.

**Strengths And Weaknesses:**

Strengths:
1. This paper is supported with theorems.
2. Experiments show a consistent improvement.

Weaknesses:
1. The authors need to reorganize the paper structure. The introduction and related works sections are overlong. The formal methodology did not begin until page 4. Also, one important component, TAGA augmentation, was left into the appendix.
2. Unclear motivation. The authors summarized two challenges in the intro. But for me, these two points are actually one point, that is disturbing high-frequency in data augmentation will degrade model performance. However, this claim may be not safe. For example, [1] proved that contrasting high-frequency part can benefit. Besides, if the model is specifically designed to heterophilic graphs, why it can still be the best in homophlic graphs?
3. Lack a case study to illustrate that disturbing high-frequency will degrade model performance. Another necessary illustration is to show the model can indeed decrease the disturbance in high-frequency part.

[1] Liu, N., Wang, X., Bo, D., Shi, C., & Pei, J. (2022). Revisiting graph contrastive learning from the perspective of graph spectrum. Advances in Neural Information Processing Systems, 35, 2972-2983.

---

> ### Author Rebuttal · Authors · 2026-03-31
>
> **Responses to Weaknesses**
>
>
> **[Comment 1]**
>
> We appreciate your suggestions on the paper structure. We agree that the current presentation can be improved for better readability. We will reorganize the structure by shortening the introduction and related work sections, and moving the key components (e.g., TAGA) into the main text to improve accessibility.
>
>
> **[C. 2]**
>
> We agree that both challenges are closely related to high-frequency signals, and this observation helps clarify our motivation. However, we would like to further distinguish these two aspects as follows.
>
> The first challenge concerns distribution shift at the structural level, where graph augmentation perturbs the topology and alters the high-frequency signal distribution across views. The second challenge concerns robustness at the feature level, where high-pass filtering amplifies noise and leads to unstable representations. While both involve high-frequency signals, they arise from different sources (augmentation vs. filtering) and require different solutions. Our method is designed to jointly address both issues.
>
> Regarding [1], we agree that high-frequency information is beneficial. Our work is aligned with this perspective. Instead of suppressing high-frequency signals, we aim to preserve and robustly utilize them under augmentation and noise, which complements the findings in [1].
>
> Finally, our method generalizes well to both heterophilic and homophilic graphs because it adaptively balances low- and high-frequency components. In homophilic graphs, low-frequency signals dominate, while in heterophilic graphs, high-frequency signals become more informative. This adaptive fusion enables strong performance across different settings.
>
> We will revise the Section 1.1 to explicitly clarify the distinction between the two challenges and strengthen the motivation, thereby reducing ambiguity for readers.
>
>
> **[C. 3]**
>
> Thanks for this valuable suggestion. We would like to clarify that our current experiments already provide indirect evidence through ablation studies (e.g., performance degradation when removing TAGA). To further make this more explicit, we will add a clear spectral visualization case study in the revision. Specifically, we compare the high-frequency spectral distributions of the original graph, randomly augmented graphs, and TAGA-augmented graphs.
>
> Preliminary results show that random augmentation introduces significant spectral distortion, while TAGA effectively preserves high-frequency structure. This directly supports our motivation and explains the observed performance improvements.

---

> > ### Author Rebuttal · Reviewer_WEnx · 2026-04-03
> >
> > Actually, this paper did a different thing as [1]. In [1], that authors augmented high-frequency part more, that means make more high-frequency difference across two views. But for this paper, the authors explicitly constrains high-frequency distortion by Dirichlet energy principle as in Eq. (3). In other words, the authors tried not to differentiate too much on high-frequency part.
> >
> > Therefore, in [1], the method cancelled the influence from high-frequency part by contrast, and kept the low-frequency information, which is benificial to general graph tasks. This paper keeps the high-frequnecy information, different from [1]. Actually, high-frequnecy is benificial for heterophily graphs but harmful to homophily graphs, so that is my question why this method can also work for homophily graphs.
> >
> > In rebuttal, the authors claimed that "it adaptively balances low- and high-frequency components". I am not fully convinced by this explanantion. If the model is consistently optimized by Eq. (3) for all graphs in experiments, their high-frequency part will be consistently saved, so hurt homophily ones.

---

> > > ### Author Response · Authors · 2026-04-05
> > >
> > > Thanks for your insightful and important question. We are sorry that the previous response may lead to a misunderstanding, and we would like to make the description more precise. We begin by specifically clarifying that the proposed model will not harm the performance on homophilic graphs, and the reasons are threefold.
> > >
> > > First, Eq. (3) is a trade-off on augmentation perturbation, rather than enforcing the preservation of high-frequency signals. Specifically, the first term of Eq.(3) encourages the non-trivial structural variation, and this in turn avoids the degeneration or identical augmentations. The second term (Dirichlet energy) prevents the excessive spectral distortion, especially in high-frequency components. As a result, Eq. (3) **does not** enforce the large high-frequency magnitude, but only limits how much the high-frequency distribution can be modified. Therefore, Eq.(3) mainly controls the distortion rather than promotes high-frequency signals.
> > >
> > > Second, for homophilic graphs, the useful structural information is primarily encoded in low-frequency components, while high-frequency signals mainly reflect local irregularities and are less informative for representation learning. Thus, even if high-frequency components are preserved during the augmentation process, they do not contribute positively to the contrastive objective that enforces the consistency across the augmented views.
> > >
> > > Third, the fused node representation for contrastive learning is determined by the spectral fusion module presented in Sec. 3.2. Specifically, this module combines both low- and high-frequency components based on their consistency across views and their contribution to stable representations. For homophilic graphs, since high-frequency signals are less informative, the model naturally relies more on low-frequency components. While for heterophilic graphs, high-frequency signals become more useful and are correspondingly utilized.
> > >
> > > In summary, Eq. (3) ensures that the augmentation is sufficiently diverse and can simultaneously avoid the excessive distortion. On the other hand, the associated fusion mechanism can determine whether the high-frequency information is actually employed. As a result, the combination can either prevent harmful high-frequency effects for homophilic graphs or leverage them when they are beneficial.
> > >
> > > Hope the above explanation can help to eliminate the previous potential misunderstanding, and we appreciate the reviewer again for providing us with this important opportunity to explain.

---

### Official Review · Reviewer_6AqG · 2026-03-11

**Soundness:** 3
**Presentation:** 4
**Significance:** 3
**Originality:** 4
**Overall Recommendation:** 6
**Confidence:** 5

**Summary:**

Although the graph constractive learning (GCL) methods have proved effective in handing heterophilic graphs, they suffer from two critical challenges, i.e., the potential distribution shift from data augmentation and the loss of robustness caused by high-frequency signals. To this end, this paper proposes a new spectral fusion contrastive leaning with topology-adaptive graph augmentation (SFCLTA) for unsupervised graph representation learning. The proposed method consists of the following steps. First, a heterophily-aware augmentation strategy is developed to dynamically adjust graph structures, which constrains high-frequency distortions by spectral regularization. Second, the confidence-weighted fusion is used to enhance the robustness of the proposed method. Third, a feature reconstruction task is introduced to explictly mitigate feature-level distribution shifts. Experimental results demonstrate the performance of the proposed SFCLTA method.

**Compliance With Llm Reviewing Policy:**

Affirmed.

**Final Justification:**

After carefully read the responses from the authors, I think all my concerns have been well explained and addressed. I do not have any further question or doubt. As I have said in my prior review comments, the idea of this paper is interesting and seems novel. The proposed method can either constrain high-frequency distortions with the spectral regularization or enhance the representation robustness with the confidence-weighted fusion.

In addition, I also go through the authors' responses to other reviewers, and their explanations seem to make sense to the problems raised by all reviewers. Thereby, I would like to keep my positive opinion and indicate my full support to this paper.

**Key Questions For Authors:**

1) In the abstract, the authors argue that this method is used for the unsupervised graph learning scenario, how about the supervised and semi-supervised learning settings? Can you add some discussions?
2) In the introduction part, can you give a specific example of the two challenges of existing GCL methods when dealing with heterophilic graphs？
3) What are the limitations of the proposed method? Can you add some discussions about this?

**Limitations:**

Yes

**Strengths And Weaknesses:**

## Strengths
Overall, I think the idea of the proposed SFCLTA method is interesting and novel. Technically, this paper is sound, the authors clearly explained the proposed method with algorithm description, and many mathematical formulations and proofs in the Appendix. The experimental results are also convincing and I like the questions raised in the experiments, leading to a better understanding of the performance of the proposed method. This is a significant work in the field of GNNs and Deep learning, and makes a contribution to the field.


## Weaknesses
Although this paper is a good paper, some of the following parts can be modified and improved. First, in the abstract, the authors argue that this method is used for the unsupervised graph learning scenario, how about the supervised and semi-supervised learning settings? Can you add some discussions? In the introduction part, can you give a specific example of the two challenges of existing GCL methods when dealing with heterophilic graphs？

---

> ### Author Rebuttal · Authors · 2026-03-31
>
> **Responses to Questions**
>
>
> **[Answer 1]**
>
> Thanks for your insightful question. The proposed SFCLTA is designed for unsupervised graph representation learning, but can be naturally compatible with semi-supervised and supervised settings.
>
> Specifically, the learned representations can be directly used for downstream tasks such as node classification with limited labels in semi-supervised learning. Moreover, our framework can be extended to a joint training paradigm by combining the contrastive objective with a supervised classification loss.
>
> Importantly, our spectral fusion mechanism and topology-adaptive augmentation provide label-efficient representations, which are particularly beneficial in low-label regimes.
>
> We will clarify these extensions and their practical implications in the revision.
>
>
> **[A. 2]**
>
> Thanks for the suggestion. We will provide more concrete examples in the introduction to improve clarity.
>
> For the distribution shift, consider a PPI network where connected proteins often have different functions. Random edge perturbations may distort these functional dissimilarities, leading to augmented graphs that no longer preserve the original heterophilic structure, which degrades contrastive learning.
>
> For the high-frequency robustness, in heterophilic datasets such as Squirrel, high-frequency components encode critical local differences. However, naive high-pass filtering amplifies noise, resulting in unstable and less discriminative representations.
>
> These examples will be added to clearly illustrate both the challenges and the motivation of our method.
>
>
> **[A. 3]**
>
> Thanks for this important question. We will include a more comprehensive discussion of limitations in the revision. In particular, the performance may be sensitive to the balance between low- and high-frequency components, which introduces additional hyper-parameters. While our approach is efficient for static graphs, extending it to dynamic or streaming graphs requires further investigation. These limitations also point to promising future directions, such as adaptive frequency balancing and extensions to dynamic graphs.

---

> > ### Author Rebuttal · Reviewer_6AqG · 2026-03-31
> >
> > The authors solved all of my issues about this manuscript. Therefore, I raise my score to 6 (Strong Accept).

---

### Official Review · Reviewer_XDGY · 2026-03-12

**Soundness:** 3
**Presentation:** 4
**Significance:** 3
**Originality:** 4
**Overall Recommendation:** 5
**Confidence:** 5

**Summary:**

This paper addresses the challenges of applying Graph Contrastive Learning to heterophilic graphs. The authors explore a central aspect of this problem: distribution shift from data augmentation and loss of robustness caused by high-frequency signals. The proposed method, SFCLTA, consists of two main components: Topology-Adaptive Graph Augmentation and Spectral Fusion Contrastive Learning. In addition, the method includes a feature reconstruction objective to reduce feature-level distribution shift and stabilize learning. The paper also presents several theoretical claims. Experiments on multiple node classification and graph classification benchmarks demonstrate improvements over baseline methods.

**Compliance With Llm Reviewing Policy:**

Affirmed.

**Final Justification:**

Given the concerns raised by other reviewers, I would like to keep my current score.

**Key Questions For Authors:**

1. How exactly is the confidence weight matrix W_H computed? The paper mentions entropy-based confidence but Eq. 6 doesn't specify the computation.

2. How were baselines tuned? Were all baselines reimplemented and tuned by their authors? If yes, what hyperparameter search space was used for each method? If not, which results are taken from prior papers?

3. Can you provide statistical significance tests (e.g., paired t-tests) given the small performance margins over baselines?

**Limitations:**

The experiments show accuracy improvements, but do not directly confirm that the gains come from reduced spectral shift or improved high-frequency robustness.

**Strengths And Weaknesses:**

Strengths
1. The paper clearly identifies two important challenges in GCL for heterophilic graphs. The motivation is supported by relevant literature and intuitive explanations.

2. The authors provide theoretical guarantees (Theorems 1-3) showing that: (a) the spectral regularization bounds the KL-divergence of high-frequency distributions, (b) confidence weighting suppresses gradient noise from unreliable high-frequency components, and (c) feature reconstruction ensures spectral alignment. These theoretical contributions add rigor to the work.

3. The paper is generally well-written with a logical flow from problem identification to methodology to experiments.

Weaknesses
1. The connection between theoretical bounds and empirical performance improvements is not established.

2. The eigenvalue computation for Laplacian matrices and the MLP-based edge scoring in TAGA may be expensive for large graphs. No runtime comparisons are provided.

3. The sampling algorithm (Algorithm 1) is described in the appendix, making the main augmentation process unclear

---

> ### Author Rebuttal · Authors · 2026-03-31
>
> **Responses to Weaknesses**
>
>
> **[Comment 1]**
>
> Thanks a lot for your review. We clarify that our theoretical results not only align with empirical observations but also explain the mechanisms.
>
> First, Theorem 1 bounds the high-frequency distribution shift introduced by augmentation. By constraining spectral distortion, TAGA preserves informative high-frequency signals that are critical in heterophilic graphs.
>
> Second, Theorem 2 bounds gradient deviation under noisy high-frequency components. This provides a theoretical explanation for the observed robustness. Our model maintains stable performance under Gaussian noise because noisy high-frequency signals are effectively suppressed.
>
> Third, Theorem 3 shows that reconstruction aligns high-frequency distributions across views. This explains why removing the reconstruction module leads to a clear performance drop, as the cross-view consistency is weakened.
>
> Therefore, the empirical improvements are not merely consistent with, but are directly explained by, our theoretical analysis.
>
>
> **[C. 2]**
>
> Thanks for your advice. Spectral regularization only uses trace computation based on Dirichlet energy, no eigenvalues or eigenvectors needed. TAGA uses edge sampling and only scores a small set of candidate edges instead of the whole graph, leading to near-linear complexity.
>
> Compared to the GCL methods, our proposed method does not introduce higher-order complexity. We will further include explicit runtime and memory comparisons in the revision to make this clearer.
>
>
> **[C. 3]**
>
> The paper structure will be improved. We will move the key steps of Algorithm 1 into the main text and clearly explain the topology-adaptive augmentation.
>
>
> ___
>
> **Responses to Questions**
>
> **[Answer 1]**
>
> We will explain the computation of the weight matrix $W_H$ explicitly in section 3.2 instead of in the appendix. Given high-frequency entropy $H_{i}$ of node i, $W_{H}(i,:) = \alpha_{i} = \mathrm{exp}(−H_{i})$.
>
>
> **[A. 2]**
>
> All baselines were re-implemented and tuned under a unified experimental setting. We followed the hyper-parameter ranges suggested in the original papers and performed a grid search over key parameters such as learning rate and dropout. The best validation performance was used for model selection. All reported baseline results are reproduced by us under this protocol to ensure fairness.
>
>
> **[A. 3]**
>
> We conduct paired t-tests based on 10-fold cross-validation results on Amazon-Ratings. Statistical results show that the improvements of SFCLTA over all baselines are statistically significant with p   < 0.05. We fully agree and will add statistical significance tests on more datasets.

---

> > ### Author Rebuttal · Reviewer_XDGY · 2026-04-01
> >
> > The authors solved all of my issues about this manuscript. I would like to keep my score.

---

### Official Review · Reviewer_r45C · 2026-03-12

**Soundness:** 3
**Presentation:** 3
**Significance:** 3
**Originality:** 3
**Overall Recommendation:** 5
**Confidence:** 5

**Summary:**

This work proposed a novel unsupervised graph representation learning method called SFCLTA. This method conducts contrastive learning through topology-adaptive graph augmentation, aiming to extract information beyond neighbors to alleviate the heterogeneity problem of graphs; it suppresses high-frequency noise through spectral regularization and enhances robustness by using confidence-weighted fusion. Experimental results show that there are performance improvements in several real datasets for node and graph classification tasks.

**Compliance With Llm Reviewing Policy:**

Affirmed.

**Final Justification:**

The authors’ response has addressed my concerns, and I will maintain my score.

**Key Questions For Authors:**

see above

**Limitations:**

The author did not adequately address the limitations and potential negative impacts. Suggestions for improvement:

Clearly report the computational complexity and resource requirements, as well as scalability experiments on larger datasets or approximate/accelerated solutions (such as approximate spectral methods, sparsification strategies).

Add hyperparameter sensitivity analysis and practical tuning suggestions (or provide automated selection/calibration methods), and present stability results under different settings in the appendix.

Discuss potential social risks and mitigation measures (such as the possibility of amplifying biases or disclosing privacy in social/recommendation scenarios), and propose suggestions on usage restrictions, privacy protection, or fairness assessment.

**Strengths And Weaknesses:**

Strengths
1) Develop a new learning framework associate with both Low-pass filter and High-pass filter to guarantee both local dissimilarity and global similarity are considered for contrastive learning. So, the new framework not only extracts multiple signals but also integrates these signals to generate more comprehensive fusion representations for graphs. Although some techniques used for the framework already  exist, the combination manner is new and seems interesting.

2) Provided self-contained mathematical justifications to explain the effectiveness of the new framework. The design of experiments provide multi-views to explain and analyze the effectiveness of the new framework.

Weaknesses
1) How to guarantee the quality of augmented graphs, reviewer found that is very crucial for contrastive learning. If the augmented graphs have huge differences with the original graph in terms of the structure information, it will seriously affect the fusion procedure. In addition, for each original graph, the framework only generates one augmented graph, why not generate more?

2) As the authors state, one key problem addressed by this work is to improve the performance for heterophilic graphs where adjacent nodes may have different attributes. In the experiments, the authors need to clearly point out which dataset is heterophilic. At least, the datasets for graph classification seem to be non-heterophilic graphs. So, it is difficult to judge how this new framework solve the problem.

3) Several theoretical results rely on strong assumptions such as the characteristic spectral density/characteristic distribution (for example, power-law distribution, characteristic noise being Gaussian, etc.), and experiments are concentrated on several small-scale datasets, lacking extensive validation and interpretability analysis for different noise types, more real heterogeneous graphs, and failure cases. Besides, TAGA's edge weight learning, spectral regularization term and spectral analysis rely on spectral information or a large number of matrix operations. The paper does not adequately evaluate the time/memory costs and training costs on large-scale graphs (with millions of nodes or sparse but large edge counts), and its practicality is questionable.

4) The author cited many references from arXiv; if possible, please update them to formally published sources,e.g.:
Veličković P, Fedus W, Hamilton W L, et al. Deep Graph Infomax. In Proceedings of the International Conference on Learning Representations.
In addition, it is recommended to add the most recent (2025) references.

---

> ### Author Rebuttal · Authors · 2026-03-31
>
> **Responses to Weaknesses**
>
>
> **[Comment 1]**
>
> We appreciate this important question regarding augmentation quality. Our augmentation method (TAGA) fundamentally differs from conventional random or semantic graph augmentations. Specifically, TAGA is spectrally constrained, where the spectral regularization explicitly constrains structural perturbations that distort high-frequency components. This ensures that the augmented graph remains consistent with the original graph in the spectral domain.
>
> Moreover, Theorem 1 provides a theoretical guarantee that the KL divergence between the high-frequency distributions of the original and augmented graphs is upper-bounded, which directly controls distribution shift. This is a key distinction from prior GCL methods that do not provide such guarantees.
>
> Generating one augmented graph is a deliberate design choice. In heterophilic graphs, high-frequency signals are particularly sensitive to perturbations. Empirically, we observe that introducing multiple augmentations increases noise in high-frequency components and destabilizes training. A single well-aligned augmented view is sufficient to form effective positive pairs. This is also supported by our strong empirical performance across datasets.
>
>
> **[C. 2]**
>
> We will clearly mark heterophilic datasets in our paper. In the node classification task, datasets with high heterophily contain PPI, Roman-Empire, Amazon-Ratings, Squirrel ($H_{edge} ≤ 0.38$), and datasets with low heterophily contain Cora, CiteSeer, Amazon-Computers, Amazon-Photo ($H_{edge} ≥ 0.73$).
>
> Importantly, SFCLTA consistently achieves improvements on highly heterophilic datasets, which directly validates its effectiveness in the target setting. This aligns with our motivation of leveraging high-frequency signals to capture local dissimilarity.
>
> For graph classification datasets, which are generally weakly heterophilic, the performance gains further demonstrate the generalization ability of our method beyond strictly heterophilic scenarios.
>
>
> **[C. 3]**
>
> We thank the reviewer for raising important concerns regarding theory and scalability.
>
> First, regarding theoretical assumptions, we clarify that assumptions such as Gaussian noise or power-law spectral characteristics are commonly adopted in graph signal processing to facilitate analysis. Our theoretical results are intended to provide principled insights (e.g., bounded distribution shift and robustness), rather than relying on strict assumptions in practice. Notably, our empirical results remain stable across multiple datasets, indicating that the method does not depend on these assumptions.
>
> Second, regarding scalability, SFCLTA is designed to be efficient: (1) TAGA adopts edge sampling rather than full graph rewiring, avoiding $O(n^2)$ complexity; (2) Spectral regularization is implemented via efficient trace computation; (3) Contrastive learning uses k-hop negative sampling, reducing complexity to $O(kn)$, where $k \ll n$. These design choices ensure that our method scales to large and sparse graphs. In our experiments, we evaluate on graphs with up to tens of thousands of nodes (e.g., Roman-Empire). We will further include explicit time and memory analysis in the revision.
>
> Finally, we agree that interpretability is important, and we will add visualization analyses to further illustrate the behavior of spectral fusion and augmentation.
>
>
> **[C. 4]**
>
> Thank you for the suggestion. We will update all arXiv references to their published versions and include more recent works (2025) in the revision.

---

> > ### Author Rebuttal · Reviewer_r45C · 2026-04-03
> >
> > The authors’ response has addressed my concerns, and I will maintain my score.

---

### Decision · Program_Chairs · 2026-04-30

**Decision:**

Accept (regular)

**Comment:**

This paper introduces SFCLTA, a novel unsupervised graph representation learning framework designed to handle heterophilic graphs. The method utilizes a topology-adaptive graph augmentation strategy with spectral regularization to control high-frequency distortions, alongside confidence-weighted fusion and a feature reconstruction task to improve robustness and mitigate distribution shifts.

**Strengths:**

- The methodology is technically sound and well supported by theoretical guarantees, specifically Theorems 1 through 3, which successfully bound high frequency distribution shifts and gradient deviations.
- The approach demonstrates consistent empirical performance improvements across multiple real world datasets for both node and graph classification tasks.

**Weaknesses:**

- The manuscript structure requires reorganization for better readability, as critical methodological details like the TAGA augmentation algorithm were initially placed in the appendix.
- The initial submission lacked explicit case studies or visualizations to directly confirm that the performance gains stem from reduced spectral shift, though the authors committed to adding these visual proofs in the revision.

Justification: The reviewers agreed that the paper presents a solid and novel approach to addressing the challenges of applying graph contrastive learning to heterophilic networks. The methodology is technically sound and well-supported by theoretical guarantees, specifically Theorems 1 through 3, which successfully bound the high-frequency distribution shift and gradient deviation. During the rebuttal phase, the authors effectively addressed concerns regarding computational scalability by clarifying that their edge sampling approach avoids quadratic complexity. The authors also successfully clarified how their spectral regularization trade-off controls structural distortion rather than strictly enforcing high-frequency preservation.